# Preeclampsia Affects Lipid Metabolism and HDL Function in Mothers and Their Offspring

**DOI:** 10.3390/antiox12040795

**Published:** 2023-03-24

**Authors:** Julia T. Stadler, Hubert Scharnagl, Christian Wadsack, Gunther Marsche

**Affiliations:** 1Division of Pharmacology, Otto Loewi Research Center for Vascular Biology, Medical University of Graz, Neue Stiftingtalstraße 6, 8010 Graz, Austria; 2Clinical Institute of Medical and Chemical Laboratory Diagnostics, Medical University of Graz, Auenbruggerplatz 15, 8036 Graz, Austria; 3Department of Obstetrics and Gynecology, Medical University of Graz, 8036 Graz, Austria; 4BioTechMed-Graz, 8010 Graz, Austria

**Keywords:** preeclampsia, HDL, cholesterol efflux, anti-oxidative capacity, HDL subclasses

## Abstract

Preeclampsia (PE) is linked to an overall increased cardiovascular risk for both the mother and child. Functional impairment of high-density lipoproteins (HDL) may contribute to the excess cardiovascular risk associated with PE. In this study, we investigated the effects of PE on maternal and neonatal lipid metabolism, and the parameters of HDL composition and function. The study cohort included 32 normotensive pregnant women, 18 women diagnosed with early-onset PE, and 14 women with late-onset PE. In mothers, early- and late-onset PE was associated with atherogenic dyslipidemia, characterized by high plasma triglycerides and low HDL-cholesterol levels. We observed a shift from large HDL to smaller HDL subclasses in early-onset PE, which was associated with an increased plasma antioxidant capacity in mothers. PE was further associated with markedly increased levels of HDL-associated apolipoprotein (apo) C-II in mothers, and linked to the triglyceride content of HDL. In neonates of early-onset PE, total cholesterol levels were increased, whereas HDL cholesterol efflux capacity was markedly reduced in neonates from late-onset PE. In conclusion, early- and late-onset PE profoundly affect maternal lipid metabolism, potentially contributing to disease manifestation and increased cardiovascular risk later in life. PE is also associated with changes in neonatal HDL composition and function, demonstrating that complications of pregnancy affect neonatal lipoprotein metabolism.

## 1. Introduction

Pre-eclampsia (PE) is a pregnancy complication that affects 4–5% of pregnancies worldwide and has serious consequences for mothers and their children [1,2]. This disorder is one of the main causes of maternal and fetal morbidity and mortality, particularly in low- and middle-income countries [3]. PE manifests after 20 weeks of gestation and is defined by the onset of hypertension (systolic ≥ 140 mm Hg, diastolic ≥ 90 mm Hg), combined with proteinuria (≥300 mg/24 h) or, in the absence of proteinuria, one of the following diagnoses: thrombocytopenia, renal insufficiency, impaired liver function, cerebral or visual problems or pulmonary oedema [3,4]. Without intervention, PE can cause significant complications and can only be cured by the delivery of the baby [5,6]. After delivery, the symptoms of PE are typically ameliorated once the blood pressure returns to pre-pregnancy levels. However, this hypertensive pregnancy disorder is also considered a major risk factor for the development of cardiovascular disease later in life for both the mother [7,8] and her child [9,10,11]. 

The major characteristics of PE comprise endothelial dysfunction, increased oxidative stress in the maternal circulation and hyperlipidemia [12,13]. Specifically, high levels of triglycerides, total cholesterol, low-density lipoprotein cholesterol (LDL-C) and reduced levels of high-density lipoprotein cholesterol (HDL-C) have been suggested to be associated with an increased risk of PE [14,15,16].

PE is thought to be driven by abnormal placentation caused by poor trophoblast differentiation and defective trophoblast invasion [17], leading to insufficient spiral artery remodeling [18]. This results in placental hypoperfusion, vascular dysfunction, and the release of anti-angiogenic factors, such as soluble fms-like tyrosine kinase 1, to the maternal circulation [19,20]. Consequently, this can lead to oxidative stress, hypertension, or even end-organ damage [6]. However, the underlying molecular mechanism leading to the development of abnormalities in the vasculature of the placenta remains poorly understood. 

PE is usually classified into two different subtypes based on the development of the disease. Early-onset PE develops before gestational week 34 and is usually more severe, while late-onset PE is diagnosed after week 34 [21].

Recent studies have shown that certain subclasses of lipoproteins, especially HDL, are involved in a number of important physiological functions, many of which are important for a healthy pregnancy [22]. HDL is now valued for its role in regulating lipid metabolism, hemostasis, immune response, inflammation, complement activation and vitamin transport [23]. HDL particles exhibit cardio-protective properties [24], including the promotion of cholesterol efflux from peripheral cells and transport to the liver for excretion [25]. Further, HDL shows anti-inflammatory [26], anti-oxidative [27] and anti-thrombotic [28] effects, including regulation of endothelial functions by promoting nitric oxide production and maintaining endothelial integrity [28]. HDL particles comprise a considerable number of proteins and are structurally and functionally heterogenous [22,29]. Apolipoprotein (apo)A-I and apoA-II stabilize HDL particles, and depict anti-oxidant properties; whereas, HDL-associated apoC-II acts as a cofactor for lipoprotein lipase, thereby regulating the hydrolysis of triglycerides. Furthermore, several enzymes are associated with HDL, such as paraoxonase-1 (PON1), a hydrolytic enzyme with a wide range of substrates and partly responsible for the anti-oxidative and anti-inflammatory properties of HDL [30]. The HDL-associated enzyme lecithin-cholesterol acyltransferase (LCAT) participates in the remodeling and maturation of HDL [31]. Functionally, LCAT is the major source of plasma-derived cholesteryl esters, and is responsible for the conversion of nascent or lipid-poor HDL into spherical HDL [32]. Disturbances in the activity of LCAT lead to altered lipoprotein metabolism, which has been described to be present and to execute crucial roles in several diseases [33,34,35]. 

In the present study, we hypothesized that maternal, but also neonatal HDL composition and function are altered in PE-affected pregnancies. As early-onset and late-onset PE are assumed to have different pathophysiological origins, we aimed to investigate the effects of these pregnancy complications on HDL metabolism and function. We characterized lipoprotein subclass distribution, compared functional metrics of HDL and investigated differences in lipid protein and apolipoprotein composition from normotensive and PE pregnancies. 

## 2. Materials and Methods

### 2.1. Recruitment and Group Characteristics

The study protocol was approved by the local ethical committee of the Medical University of Graz. Women without pregnancy complications and normal blood pressure levels (*n* = 32), early-onset PE (*n* = 18) and late-onset PE (*n* = 14) were recruited at the time of delivery, and gave informed written consent (26-333 ex 13/14). Women with multiple births were excluded from the study. PE was defined as new-onset hypertension above 140/90 mm Hg occurring after 20 weeks of gestation, and one or multiple of the following emerging conditions: proteinuria, renal insufficiency, thrombocytopenia, compromised liver function, pulmonary edema, uteroplacental dysfunction or neurologic complications. The occurrence of PE was defined as early-onset if it was detected before 34 weeks gestation, or late-onset if it was detected after 34 weeks of gestation [2]. Healthy participants were selected based on normal blood pressure and the absence of medical complications during pregnancy, and were matched for age and pre-pregnancy BMI of the PE groups. Venous blood from pregnant women at term was collected before delivery, while corresponding umbilical cord blood was collected no later than 10 min after delivery. EDTA plasma was isolated by centrifugation at 3500 rpm for 10 min at 4 °C, and stored at −80 °C until further analysis. Blood pressure, C-reactive protein levels and platelet count were assessed among all participants, whereas the concentration of soluble fms-like tyrosine kinase-1 (sFlt-1), placental growth factor (PlGF) and liver markers were only measured in PE women. 

### 2.2. ApoB-Depleted Plasma and Plasma Lipids

ApoB-depleted plasma was prepared by the addition of 40 μL polyethylenglycol (P1458, Sigma-Aldirch, Darmstadt, Germany) (20% in 200 mmol/L glycine buffer) to 100 μL plasma, followed by gentle mixing. Plasma samples were incubated for 20 min at room temperature. After centrifugation at 10,000 rpm for 30 min at 4 °C, the supernatant was collected and stored at −70 °C until use. Enzymatic photometric transmission measurement (Roche Diagnostics, Mannheim, Germany) was used to measure plasma lipids, such as total cholesterol, triglycerides and HDL-C. LDL-cholesterol concentrations were calculated using the Friedewald’s formula [36].

### 2.3. HDL-Associated Apolipoproteins and Lipids

HDL-associated apoA-I, apoA-II, apoC-II, apoC-III and apoE were determined by immunoturbidimetry in apoB-depleted plasma [37]. HDL-associated lipids, including cholesterol, phospholipids and triglycerides, were determined by enzymatic techniques, as previously described [38]. Cholesteryl-ester levels were calculated as the difference between total cholesterol and free cholesterol, measured in apoB-depleted plasma. As previously described, all lipoprotein analyses were performed on an Olympus AU680 analyzer (Beckman Coulter, Brea, CA, USA) [37]. To determine HDL particle composition, apolipoproteins/lipids were corrected for total HDL protein.

### 2.4. HDL Subclass Distribution

HDL subfractions were determined using the Lipoprint© System (48-9002, Quantimetrix, CA, USA), according to the manufacturer’s instructions. This system separates HDL subclasses from human plasma on the basis of size, using preloaded gel tubes for HDL determinations [39]. In brief, 25 μL of plasma of each patient was loaded into polyacrylamide gel tubes, along with 300 μL loading gel solution containing a lipophilic dye. The tubes were photopolymerized at room temperature for 30 min. Electrophoresis with tubes containing plasma samples, along with the manufacturer’s quality controls, was performed at a constant of 3 mA/tube for 50 min. Subfraction bands were scanned and identified by their mobility (Rf) using very-low-density lipoprotein and low-density lipoprotein as the starting (Rf 0.0) and albumin as the ending (Rf 1.0) reference points. Ten HDL subfractions were identified and grouped into three major classes: large (HDL1 to HDL3), intermediate (HDL4 to HDL7) and small (HDL8 to HDL10) subfractions [39]. 

### 2.5. Cholesterol Efflux Capacity Assay of Apob-Depleted Plasma

Evaluation of cholesterol efflux capacity was performed as described elsewhere [40,41]. J774.2 cells (Sigma-Aldrich, Darmstadt, Germany) were cultured in DMEM (Life Technologies, Carlsbad, CA, USA), containing 10% fetal bovine serum and 1% penicillin/streptomycin. In each well, 300,000 cells were seeded on 48-well plates (Greiner Bio-One, Kremsmünster, Austria), cultured for 24 h and labeled with 0.5 µCi/mL radiolabeled [^3^H]-cholesterol (ART0255, Hartmann Analytic, Braunschweig, Germany) in DMEM containing 2% BSA, in the presence of 0.3 mM 8-(4-chlorophenylthio)-cyclic adenosine monophosphate (c3912, Sigma-Aldrich, Darmstadt, Germany) overnight. Cyclic adenosine monophosphate was used for the upregulation of ATP-binding cassette transporter A1. After 18 h, the cells were rinsed with DMEM (serum-free) and equilibrated with DMEM (serum-free) containing 2 mg/mL bovine serum albumin (Sigma-Aldrich, Darmstadt, Germany) for 2 h. Cells were then incubated with 2.8% apoB-depleted plasma for 3 h to determine [^3^H]-cholesterol efflux. The cholesterol efflux capacity was expressed as the radioactivity in the medium in relation to the total radioactivity in medium and cells. All steps were performed in the presence of 2 µg/mL acyl-coenzyme A cholesterol acyltransferase inhibitor Sandoz 58-035 (Sigma-Aldrich, Darmstadt, Germany).

### 2.6. Arylesterase Activity of PON1

As described elsewhere, the arylesterase activity of HDL-associated paraoxonase was evaluated by a photometric assay, using phenylacetate (10873, Sigma-Aldrich, Darmstadt, Germany) as substrate [42]. 

### 2.7. Anti-Oxidative Capacity

Plasma anti-oxidative activity was determined using fluorometric assay, as previously described [43]. The ability of apoB-depleted plasma samples to inhibit dihydrorhodamine (Cay85100-5, Biomol, Hamburg, Germany) oxidation was monitored.

### 2.8. LCAT Activity

Plasma lecithin-cholesterol acyltransferase (LCAT) activity was determined using a commercially available kit (MAK107, Merck, Darmstadt, Germany), according to the manufacturer’s instructions. Briefly, plasma samples were incubated with the LCAT substrate at 37 °C for a period of 4 h. The fluorescent substrate emits fluorescence at a wavelength of 470 nm. When the substrate is hydrolyzed by LCAT, a monomer is released, which emits fluorescence at 390 nm. LCAT activity was assessed over time and expressed as the change of 470/390 nm emission intensity.

### 2.9. VLDL and LDL Subclass Distribution

Plasma levels of very-low-density lipoprotein (VLDL), intermediate-density lipoprotein (IDL) and distribution of LDL subclasses were assessed using the Lipoprint© System (48-7002, Quantimetrix Corp., Redondo Beach, CA, USA) according to the manufacturer’s instructions. Plasma samples were loaded on gel tubes and mixed with 200 µL of Lipoprint loading gel, containing a lipophilic dye. The dye proportionally binds to lipids in the plasma. Gel tubes were photopolymerized for 30 min. Electrophoresis was performed for 60 min at 3 mA per gel tube and 500 V maximum. Gel tubes were scanned and analyzed using the Lipoware software (Lipoware HDL Research LW03-v.16-134), after a rest period of 30 min.

### 2.10. Statistical Analysis

Differences between the two groups were analyzed using an ANOVA or Kruskal–Wallis test, followed by multiple comparisons corrected according to Bonferroni. Individual data are depicted on top of boxplots, showing the median and interquartile range, as well as the minimum and maximum values. Correlations were assessed using a Spearman’s correlation coefficient rho, due to the skewed nature of many of the parameters. Statistical analyses were performed using GraphPad Prism (Version 9.5.0, GraphPad Software, San Diego, CA, USA) and SPSS Statistics (Version 26, Armonk, NY, USA: IBM Corp).

## 3. Results

### 3.1. Maternal and Fetal Characteristics of the Study Population

In this explorative study, we included 32 normal pregnancies, 18 early-onset PE and 14 late-onset PE cases, of which maternal and cord blood samples were collected. The control cohort had normal blood pressure levels and no incidence of pregnancy complications. Some of the risk factors for developing PE include nulliparity, high pre-pregnancy BMI, age, chronic hypertension, pre-gestational diabetes or renal disease [2]. As it has already been shown that BMI impacts HDL function in mothers and neonates [44], our participants were matched for age and pre-pregnancy BMI to exclude possible confounding factors. Clinical characteristics are shown in Table 1. 

As expected, the gestational age was lower in the early-onset PE group, compared to normal pregnancy, while it was not different in late-onset PE patients. Significant differences in both PE groups were observed in systolic and diastolic blood pressure levels compared to the control group. The study groups did not show differences in the levels of CRP, platelet count or mode of delivery. A significant difference in the levels of sflt-1 and PlGF was observed between early-onset and late-onset PE, whereas the concentration of uric acid and liver markers showed no differences.

Placental weight, as well as the weight of the neonate, differed significantly between normal pregnancy and PE pregnancies, while there was no difference in fetal sex. 

### 3.2. Preeclampsia Is Associated with Altered Plasma Lipid Levels in Mothers and Offspring

One of the risk factors and characteristics of PE is maternal hyperlipidemia [45,46]. In our study cohort, we observed that early- and late-onset PE was associated with atherogenic dyslipidemia in mothers, characterized by high triglycerides and low HDL-C levels. Total cholesterol and non-HDL-C levels did not differ between the maternal study groups (Figure 1A). In contrast, in cord blood, total cholesterol and non-HDL-C levels were elevated in early-onset PE compared with offspring of normal pregnancies, whereas triglyceride and HDL-C levels were not significantly different (Figure 1B).

### 3.3. Preeclampsia-Related Changes of HDL-Associated Apolipoprotein and Lipid Composition

The particle composition of HDL critically determines functionality [47]. Since we observed differences in the quantity of HDL-cholesterol levels, we determined the relative abundance of major apolipoproteins of HDL by calculating the ratio between apolipoproteins and total protein content of HDL. We observed a higher abundance of HDL-associated apoC-II in both maternal PE groups (Figure 2A). ApoC-II is a cofactor of lipoprotein lipase, promoting the hydrolysis of triglycerides [48].

No differences in the relative abundance of apolipoproteins between maternal normal pregnancy and PE were observed for apoA-I, apoA-II, apoC-III, apoE and cord blood apolipoproteins (Figure 2B).

We next assessed the abundance of major lipid constituents of HDL. To determine the HDL composition, lipid levels were measured in apoB-depleted plasma and corrected for the total protein content of HDL. While we did not find any differences between maternal PE groups and the control (Figure 3A), a significant increase in HDL-associated triglycerides in the cord blood of early-onset PE was observed (Figure 3B).

Analyses revealed that maternal HDL apoC-II levels were inversely correlated with HDL triglycerides (r_S_ = −0.287, *p* = 0.026), consistent with the concept that apoC-II is a cofactor of lipoprotein lipase [48].

### 3.4. Preeclampsia Affects Maternal HDL Subclass Distribution

HDL particles are heterogeneous in structure and composition, providing the basis for their functional variability. HDL particles can be divided into large and cholesterol-rich HDLs, and protein-rich and denser small HDLs, which also vary in their protective properties [49,50].

We next determined HDL subclass distribution. By using the Quantimetrix Lipoprint© system, we determined 10 different HDL subclasses based on their size, which were then grouped into large, intermediate and small HDL subclasses. Of particular interest, maternal early-onset PE group showed reduced levels of large and cholesterol-rich HDL particles, whereas small HDLs were increased when compared to normotensive controls (Figure 4A). No differences in HDL subclass distribution were observed in the cord blood of PE pregnancies (Figure 4B).

### 3.5. Effect of Preeclampsia on LCAT Activity in Mother and Child

LCAT is critically involved in HDL metabolism, catalyzing an important step in HDL maturation [49]. This enzyme esterifies free cholesterol to cholesteryl-ester, which converts nascent HDL into the mature and larger spherical form [51]. Given the observed differences in the distribution of HDL subclasses, we were interested in whether PE affects the enzyme activity of LCAT. Interestingly, we observed that PE had no effect on LCAT activity in maternal plasma, despite the revealed shift in HDL subclass distribution. However, we observed a reduction in LCAT activity in the cord blood plasma of early-onset PE neonates, although HDL subclass distribution in the cord blood was unaltered (Figure 5).

### 3.6. Effects of Preeclampsia on Parameters of HDL Function

HDL particles are well known to be athero-protective by promoting reverse cholesterol transport [41]. In addition, HDL particles show anti-inflammatory [26] and anti-oxidative properties [27]. In our experiments, apoB-depleted plasma (containing all HDL subclasses, but no apoB-containing lipoproteins) of the mothers and the cord blood was used to measure functional metrics of HDL. HDL cholesterol efflux capacity was determined using a well-established cell-based assay [25,52]. Of particular interest, HDL cholesterol efflux capacity was significantly reduced in the cord blood of the late-onset PE group (Figure 6B), while we observed no changes in maternal samples (Figure 6A). Paraoxonase 1 (PON1) is a HDL-associated antioxidant and an anti-inflammatory enzyme [30]. We did not detect PE-associated changes in the arylesterase activity of PON1 in either maternal or cord blood (Figure 6C,D).

We next examined the total anti-oxidative capacity of apoB-depleted plasma by determining the ability of plasma to inhibit free-radical-induced oxidation of the fluorescent dye dihydrorhodamine [53]. Against the expectation, early-onset maternal PE was associated with increased plasma antioxidant capacity (Figure 6E), whereas the cord plasma antioxidant capacity was not altered (Figure 6F). Correlation analysis further revealed that the anti-oxidative capacity of plasma correlated with the shift in HDL subclass distribution to small HDL particles in maternal early-onset PE (r_S_ = 0.301, *p* = 0.010). Moreover, we found a significant correlation between plasma uric acid and anti-oxidative capacity (r_S_ = 0.448, *p* = 0.011). Plasma uric acid is commonly elevated in subjects with impaired kidney function [54] and is a powerful antioxidant [55].

### 3.7. Association of PE with Alterations in Subclasses of Triglyceride-Rich Lipoproteins

Women affected by PE during pregnancy have a profound increased risk for cardiovascular complications later in life [56]. Certain LDL subclasses are strongly associated with cardiovascular risk [57,58,59]. We next assessed the distribution of low-density lipoproteins by using the Quantimetrix Lipoprint© system. As the abundance of LDL is much lower in cord blood than in adults [60], we were not able assess LDL subclasses in offspring samples. Of particular interest, our analyses revealed a decrease of intermediate-density lipoprotein (IDL)-C in early-onset PE and a trend (*p* = 0.088) for reduced levels in late-onset PE, while IDL-A was significantly increased in both PE groups (Figure 7). Distribution of large LDL subclasses was higher in late-onset PE, while no differences were observed for intermediate or small LDL subclasses.

## 4. Discussion

Lipoproteins are most commonly studied for their role in transporting lipids and the maintenance of tissue lipid homeostasis, and are therefore usually defined by measures of triglyceride and cholesterol content. Recent studies have shown that primarily HDLs are involved in a number of critical physiological functions, many of which are involved in ensuring a healthy pregnancy [22]. In this exploratory study, we observed PE-associated changes in lipoprotein metabolism, affecting both the mothers and their offspring. These changes lead to altered biological activity of HDL particles and changes in triglyceride-rich lipoproteins, which could contribute to the manifestation of this pregnancy complication and increased morbidity and mortality.

### 4.1. PE-Associated Alterations in Lipid Metabolism and HDL Function in Mothers

A few studies have reported that maternal hyperlipidemia is associated with the risk of PE [46,61]. Moreover, a meta-analysis revealed that PE is linked to elevated total cholesterol, non-HDL-C and, particularly, triglycerides, regardless of gestational age at the time of blood sampling [46]. In this study, we confirmed the association of plasma triglycerides with an observed increase of about 30% in early-onset PE and late-onset PE, when compared with the normotensive control group. In line with our observed results, lower levels of HDL-C in the third trimester have also been previously reported in PE [46,62,63].

Strong evidence is accumulating that hypertriglyceridemia is associated with endothelial dysfunction. Hypertriglyceridemia is accompanied by an increase in free fatty acids, which are then increasingly taken up by endothelial cells and further esterified to triglycerides [64]. The accumulation of triglycerides can harm endothelial cells and may contribute to endothelial dysfunction in PE [65]. The combination of hyperlipidemia with increased oxidative stress found in preeclamptic mothers could lead to the accumulation of oxidized lipids in the arterial wall, promoting inflammation and cardiovascular disease.

Metabolic characteristics, besides hypertriglyceridemia in PE, are hyperuricemia, hyperinsulinemia and low levels of large HDL (HDL2) particles, which are similar to the main features of insulin resistance [66]. Consistent with these previous findings, we observed lower HDL-cholesterol levels in mothers and a shift in HDL subclass distribution from large to small subclasses in early-onset PE. These changes in HDL subclass distribution might be explained by a reported increased activity of hepatic lipase in PE when compared to normotensive pregnancy [67]. Additionally in good agreement with previous reports, we observed that early- and late-onset PE were associated with markedly increased plasma triglyceride levels.

To determine whether PE is associated with alterations in HDL composition, which strongly determines HDL-protective functions [23,68,69,70], we evaluated the distribution of the most abundant apolipoproteins and the concentration of HDL-associated lipids.

In preeclamptic mothers, the HDL content of apoC-II was increased. Moreover, the HDL content of apoC-II was inversely correlated with concentration of HDL triglycerides, consistent with the fact that apoC-II is a cofactor of lipoprotein lipase, the main enzyme promoting hydrolysis of triglycerides in plasma [71].

Of particular interest, analyses of LDL subclass distribution revealed a lower percentage of IDL-C, the first intermediate particle formed after hydrolysis of VLDL in PE, while IDL-A and the large LDL subclasses were increased.

In contrast to our results, earlier studies have shown an increase in small atherogenic LDL particles and a decrease in the large buoyant LDL [67]. However, we detected no differences in small LDL, similar to a previous study [72].

One major characteristic of PE is increased oxidative stress in the maternal circulation [5,73], therefore, we investigated the effects on plasma anti-oxidative capacity. We observed that maternal anti-oxidative activity of plasma in the early-onset PE group was markedly increased. The higher anti-oxidative capacity related to hypertension during pregnancy might indicate a compensatory mechanism in response to increased oxidative stress in the circulation. Interestingly, we observed that plasma anti-oxidative capacity correlated with the PE-associated shift from large HDL to small HDL subclasses in early-onset PE. This might be explained, at least in part, by the fact that small, dense HDLs are known to exhibit potent anti-oxidant activity, which may arise from synergy in the inactivation of oxidized lipids by enzymatic and nonenzymatic mechanisms [74]. Moreover, we found a significant correlation between uric acid [54] and anti-oxidative capacity. Plasma uric acid is commonly elevated in subjects with impaired kidney function [54] and is a powerful antioxidant and scavenger of singlet oxygen and radicals [55,75]. Uric acid and other hydrophilic antioxidants could explain the relationship between plasma antioxidant capacity and markers of renal dysfunction [76]. However further studies are warranted to underline this, so far speculative, hypothesis. We further examined the activity of the HDL-associated PON1, an antioxidant and anti-inflammatory enzyme [30]. We found that PE did not affect PON1 activity. This is in contrast to some previous studies reporting decreased PON1 activity in mothers diagnosed with PE [77,78], but these studies used methods of measuring PON1 activity other than assessing arylesterase activity in apoB-depleted plasma.

### 4.2. PE-Related Alterations in Lipid Metabolism and HDL Function in Neonates

In contrast to previous studies that focused mainly on PE-affected mothers, we also emphasized measurements of HDL metabolism, composition and function in the corresponding cord blood of the offspring. The availability of many substrates for the fetus depends on their concentration in the maternal circulation and the extent to which they are transported across the placenta [79,80]. Therefore, it is also reasonable to assume that lipoprotein metabolism in PE is altered not only in the affected mothers, but also in the offspring.

We observed that total cholesterol, as well as non-HDL-C, were profoundly increased in neonates of early-onset PE, while triglyceride levels and HDL-C were unaltered. These results of lipid measurements in PE cord blood are in line with a previous study [81]. However, it has to be noted that in our study, these changes were only seen in early-onset PE, which represents the more severe type. Interestingly, we observed no differences in the composition of apolipoproteins of cord-blood-derived HDL. In line with previous reports [82], we observed that apoE levels in HDL from neonates were more than three times higher than in HDL from adults. Since apoE binds with high affinity to the LDL receptor, it appears that the primary function of apoE-enriched neonatal HDL may be cholesterol transport to tissues, as is carried out in adults by LDL [82].

We were further interested in whether PE is associated with changes in protective functions of HDL in neonates. ApoB-depleted plasma was used to assess the cholesterol efflux capacity of HDL, an anti-atherogenic property of HDL, which has been shown to be inversely correlated with coronary artery disease, independent of HDL-cholesterol concentrations [41]. While our analyses did not reveal PE-associated changes in maternal samples, cholesterol efflux capacity of HDL in neonates of late-onset PE pregnancies was profoundly reduced. Of particular interest, we have previously observed a reduction in HDL cholesterol efflux capacity in neonates of pregnancies affected by gestational hypertension, whereas no significant changes were seen in mothers [83]. Similar to our results, a previous study reported a PE-associated decrease in ABCA1-mediated HDL cholesterol efflux capacity in neonates; however, in that study, an increased total cholesterol efflux capacity of maternal plasma was observed [84].

We observed that neonates with early-onset PE depicted reduced LCAT activity, but this was not associated with changes in HDL apolipoprotein levels or HDL subclass distribution. LCAT is a key enzyme involved in the remodelling of HDL by esterifying free cholesterol on HDL surface, which leads to particle maturation [85].

Moreover, we observed that in neonates the anti-oxidative activity of plasma, as well as PON1 activity, were not significantly altered, although maternal plasma anti-oxidant activity was significantly increased in the early-onset PE group.

Some limitations should be mentioned. The control group with normal pregnancy differed from the group with early-onset PE in gestational age, accompanied by lower placental and fetal weights. However, the group with late-onset PE did not differ in gestational age, but showed alterations similar to those of the group with early-onset PE. Nevertheless, we cannot exclude the possibility that premature birth affected the observed changes. Moreover, our study is exploratory and the design is limited to being correlative in nature, not permitting causal inference. Further studies in larger cohorts are needed to confirm our results and draw firm conclusions.

## 5. Conclusions

In this study, we demonstrated that PE is associated with marked changes in maternal lipid metabolism. Of particular interest, PE also was associated with changes in neonatal HDL composition and function, demonstrating that complications of pregnancy affect neonatal lipoprotein metabolism. Despite the presumed different origin of early- and late-onset PE, we observed similar differences in maternal plasma lipid levels and HDL composition.

Notably, early-onset PE led to a shift in HDL subclass distribution from large to smaller particles, associated with an increase in plasma anti-oxidative capacity. Moreover, neonates of mothers affected by late-onset PE showed markedly reduced HDL cholesterol efflux capacity, whereas early-onset PE depicted a decreased LCAT activity. In conclusion, our results suggest that early-onset, as well as late-onset, PE affect maternal and neonatal lipid metabolism, potentially contributing to disease manifestation and increased cardiovascular risk later in life. However, larger studies are needed to confirm our results and investigate whether these changes persist after the birth of the child.

## Figures and Tables

**Figure 1 antioxidants-12-00795-f001:**
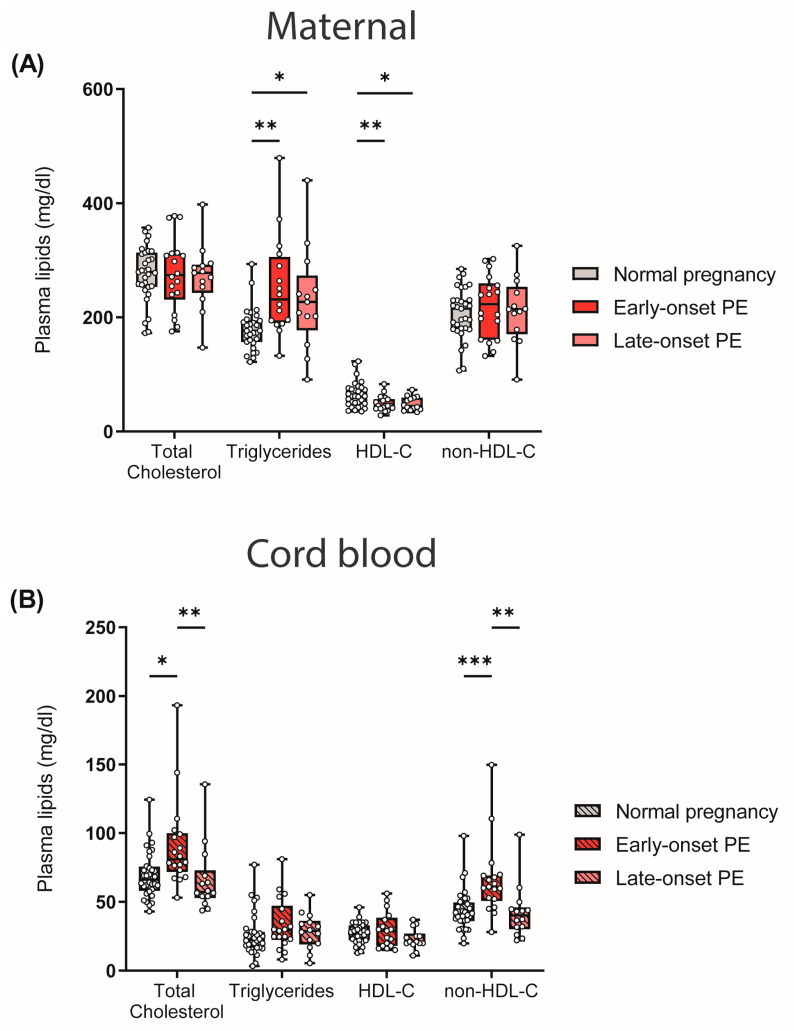
Differences between plasma lipid levels in women with normal pregnancy, early-onset PE and late-onset PE, and corresponding offspring: (**A**) Plasma total cholesterol levels, triglycerides, HDL-C and non-HDL-C of mothers and (**B**) neonates. Individual data are presented on top of boxplots, showing the median, interquartile range, and minimum and maximum values. Differences between the groups were analyzed by the Kruskal–Wallis test, followed by multiple comparisons, corrected according to Bonferroni. * *p* < 0.05, ** *p* < 0.01, *** *p* < 0.001. (Maternal: early-onset PE, *n* = 17; late-onset PE, *n* = 13; normal pregnancy, *n* = 31; cord blood: early-onset PE, *n* = 17; late-onset PE, *n* = 13; normal pregnancy, *n* = 32).

**Figure 2 antioxidants-12-00795-f002:**
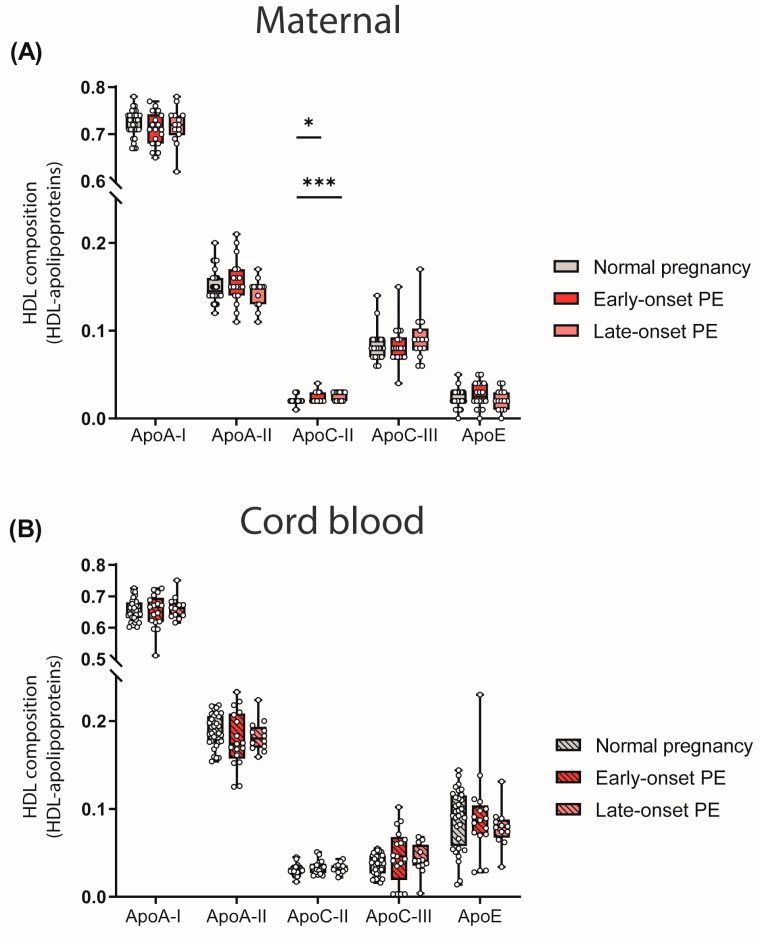
HDL composition: HDL-associated apolipoproteins in mothers with normal pregnancy, early-onset PE and late-onset PE (**A**) and in their offspring (**B**). Individual data are presented on top of boxplots, showing the median, interquartile range, and minimum and maximum values. Differences between the groups were analyzed by the Kruskal–Wallis test, followed by multiple comparisons, corrected according to Bonferroni. * *p* < 0.05, *** *p* < 0.001. (Maternal: early-onset PE, *n* = 18; late-onset PE, *n* = 14; normal pregnancy, *n* = 32; cord blood: early-onset PE, *n* = 17; late-onset PE, *n* = 13; normal pregnancy, *n* = 32).

**Figure 3 antioxidants-12-00795-f003:**
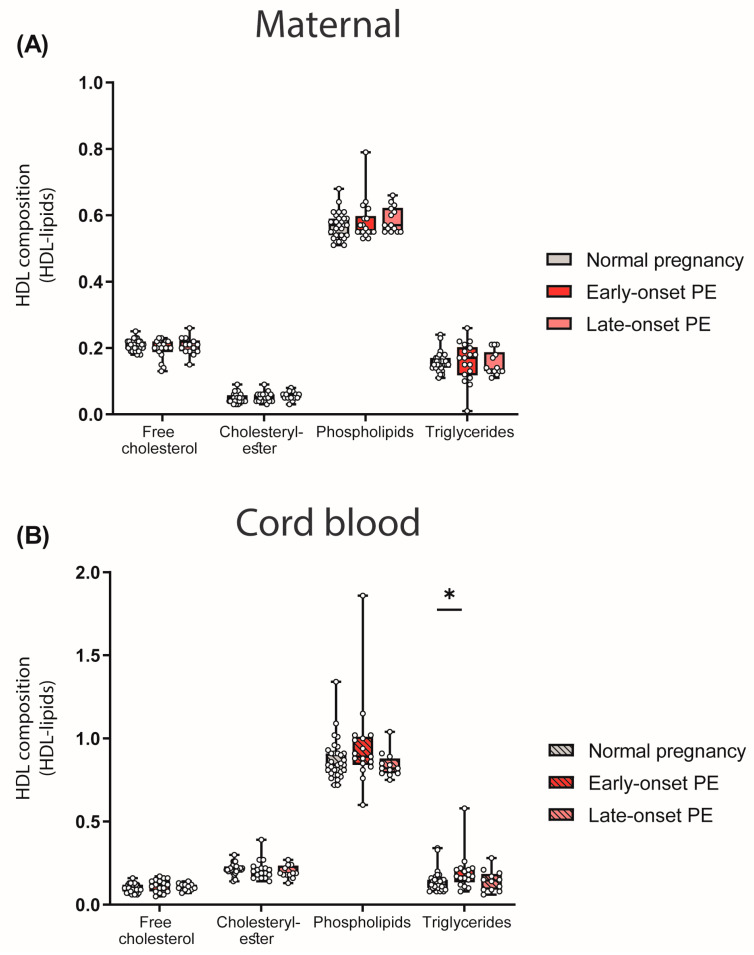
Lipid composition of HDL particles in mothers with normal pregnancy, early-onset PE and late-onset PE (**A**) and their offspring (**B**). Individual data are presented on top of boxplots, showing the median, interquartile range, and minimum and maximum values. Differences between the groups were analyzed by the Kruskal–Wallis test, followed by multiple comparisons, corrected according to Bonferroni. * *p* < 0.05. (Maternal: early-onset PE, *n* = 18; late-onset PE, *n* = 14; normal pregnancy, *n* = 32; cord blood: early-onset PE, *n* = 17; late-onset PE, *n* = 13; normal pregnancy, *n* = 32).

**Figure 4 antioxidants-12-00795-f004:**
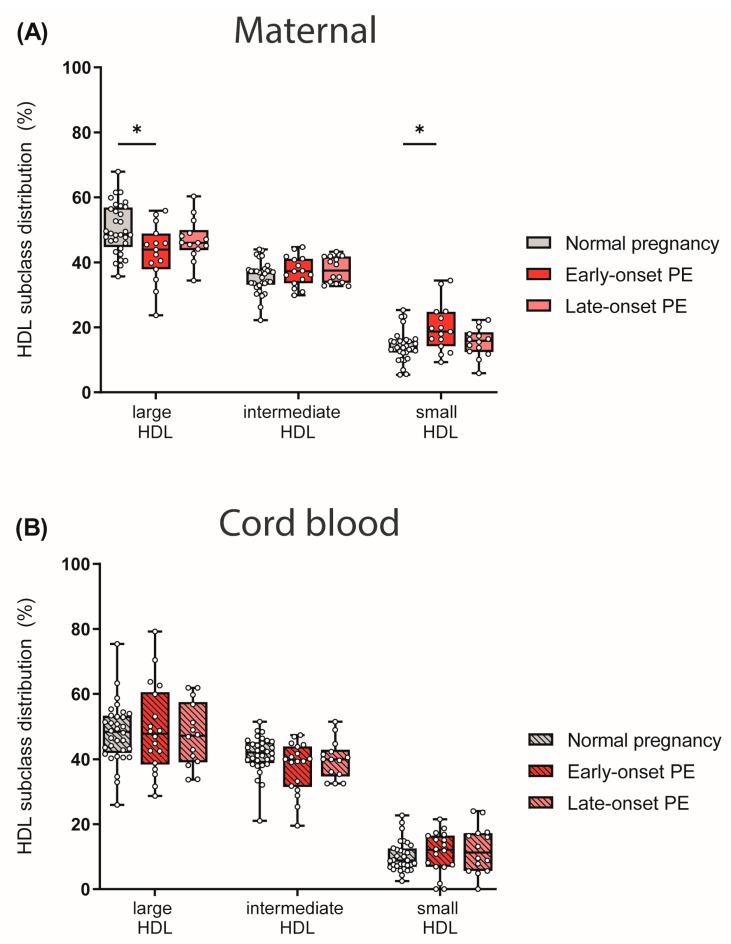
The distribution of HDL subclasses in mothers (**A**) with normal pregnancy, early-onset PE, and late-onset PE and in their offspring (**B**) was investigated. HDL subclasses were determined using the Quantimetrix Lipoprint© system. Individual data are presented on top of boxplots, showing the median, interquartile range, and minimum and maximum levels. Differences between the groups were analyzed by the Kruskal–Wallis test, followed by multiple comparisons, corrected according to Bonferroni. * *p* < 0.05. (Maternal: early-onset PE, *n* = 17; late-onset PE, *n* = 14; normal pregnancy, *n* = 32; cord blood: early-onset PE, *n* = 18; late-onset PE, *n* = 14; normal pregnancy, *n* = 32).

**Figure 5 antioxidants-12-00795-f005:**
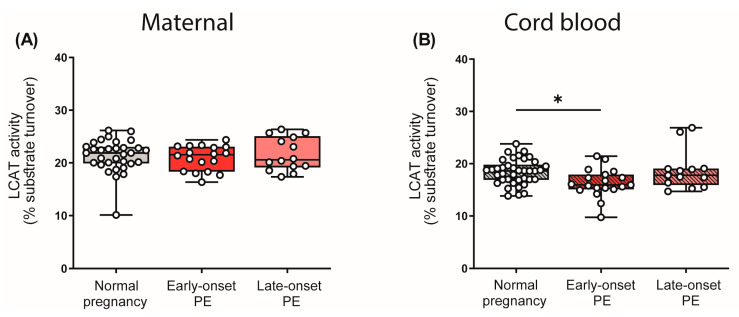
Enzyme activity of lecithin-cholesterol acyltransferase (LCAT) in the study cohort. The activity of LCAT was assessed in maternal plasma (**A**) and in the offspring (**B**). Individual data are presented on top of boxplots, showing the median, interquartile range, and minimum and maximum levels. Differences between the groups were analyzed by the Kruskal–Wallis test, followed by multiple comparisons, corrected according to Bonferroni. * *p* < 0.05. (Maternal: early-onset PE, *n* = 17; late-onset PE, *n* = 14; normal pregnancy, *n* = 32; cord blood: early-onset PE, *n* = 18; late-onset PE, *n* = 14; normal pregnancy, *n* = 32).

**Figure 6 antioxidants-12-00795-f006:**
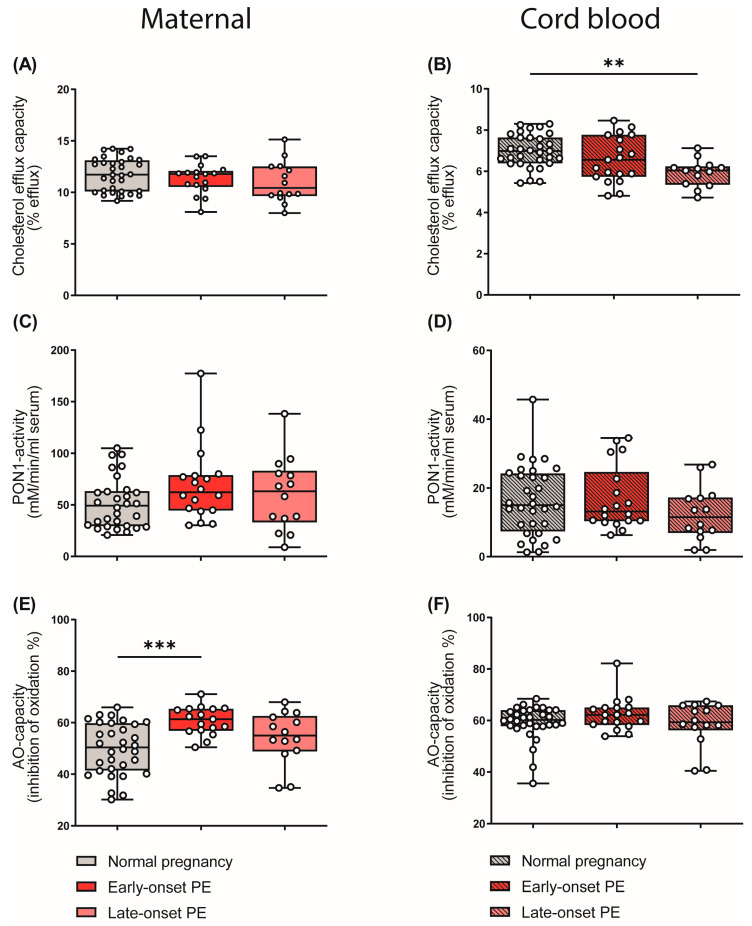
Differences in parameters of HDL function and plasma anti-oxidative capacity of mothers with normal pregnancy, early-onset PE, and late-onset PE, and their offspring. ApoB-depleted plasma was used to assess cholesterol efflux capacity (**A**,**B**) and the activity of HDL-associated paraoxonase-1 (PON1) (**C**,**D**) in mothers and offspring. The anti-oxidative capacity of plasma was assessed by measuring the inhibition of oxidation of a substrate (**E**,**F**). Individual data are presented on top of boxplots, showing the median, interquartile range, and minimum and maximum levels. Differences between the groups were analyzed by the Kruskal–Wallis test, followed by multiple comparisons, corrected according to Bonferroni. ** *p* < 0.01, *** *p* < 0.001. (Maternal: early-onset PE, *n* = 18; late-onset PE, *n* = 14; normal pregnancy, *n* = 32; cord blood: early-onset PE, *n* = 17; late-onset PE, *n* = 13; normal pregnancy, *n* = 32).

**Figure 7 antioxidants-12-00795-f007:**
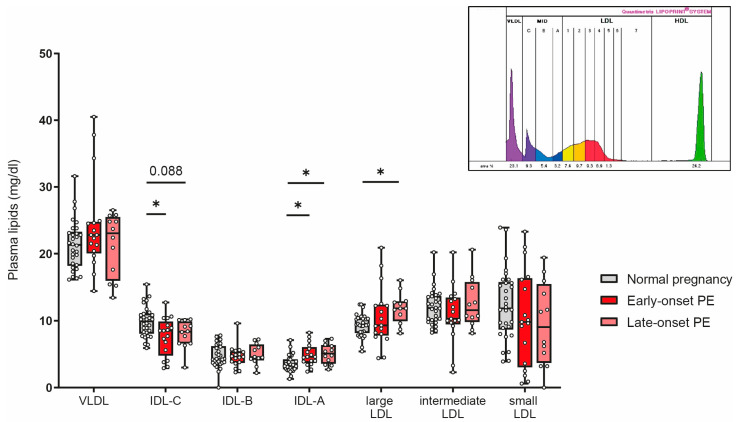
The distribution of low-density subclasses in mothers with normal pregnancy, early-onset PE, and late-onset PE was investigated. LDL subclasses were assessed using the Quantimetrix Lipoprint© system. An example of a graph of low-density lipoprotein subclass distribution in a pregnant women is shown in the insert. Individual data are presented on top of boxplots, showing the median, interquartile range, and minimum and maximum levels. Differences between the groups were analyzed by the Kruskal–Wallis test, followed by multiple comparisons, corrected according to Bonferroni. * *p* < 0.05. (Maternal: early-onset PE, *n* = 17; late-onset PE, *n* = 12; normal pregnancy, *n* = 30).

**Table 1 antioxidants-12-00795-t001:** Maternal and fetal characteristics of the study population.

Maternal Characteristics	Normal Pregnancy	Early-Onset PE	*p*-Value	Late-Onset PE	*p*-Value
Number of matched samples	32	18		14	
Maternal age (years)	30 (27–33)	34 (28–37)	ns	31 (30–36)	ns
Pre-pregnancy BMI (kg/m^2^)	22.5 (20.7–28.3)	25.4 (22.0–27.4)	ns	26.5 (22.8–29.3)	ns
Gestational age (weeks)	38.8 (38.1–39.1)	33.6 (31.7–34.3) †	<0.001	37.2 (35.5–37.9)	ns
Mode of delivery (% C-section)	88	100	ns	72	ns
Systolic blood pressure (mmHg)	117 (110–122)	163 (156–185)	<0.001	153 (143–169)	<0.001
Diastolic blood pressure (mmHg)	74 (66–79)	107 (100–115)	<0.001	101 (94–108)	<0.001
CRP (mg/mL)	4.8 (2.5–7.9)	5.1 (3.4–19.9)	ns	7.0 (2.5–17.1)	ns
Sflt-1 (pg/mL)	-	15,721 (12,184–21,500) †	-	9542 (8125–154,001)	-
PLGF (pg/mL)	-	44.5 (29.1–56.8) †	-	67.3 (36.9–86.6)	-
Platelets	212 (174–244)	202 (150–239)	ns	189 (140–220)	ns
Uric acid (mg/dL)	-	6.2 (5.5–7.0)	-	5.9 (5.1–7.1)	-
ALT (U/L)	-	30.0 (21.8–44.3)	-	24.5 (16.8–38.8)	-
AST (U/L)	-	25.0 (18.8–51.0) †	-	13.5 (8.8–45.8)	-
**Fetal characteristics**					
Sex (% female)	59	50	ns	57	ns
Weight at birth (g)	3230 (2951–3683)	1660 (1438–2021)	<0.001	2545 (2303–3185)	0.054
Placenta weight (g)	615 (563–665)	360 (320–455)	<0.001	495 (448–533)	0.005

Results are presented as the median (Q1–Q3) or as a relative abundance (%). Differences between normal pregnancy and early- as well as late-onset PE were calculated using the Kruskal–Wallis test, followed by multiple comparisons, corrected according to Bonferroni. BMI, body mass index; CRP, c-reactive protein; Sflt-1, soluble fms-like tyrosine kinase-1; PlGF, placental growth factor; ALT, alanine aminotransferase; AST, aspartate transaminase. † represents a significant difference between early-onset PE and late-onset PE.

## Data Availability

Data are contained within the article.

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
