# Peer review of "Preeclampsia Affects Lipid Metabolism and HDL Function in Mothers and Their Offspring"

_antioxidants, 2023, doi:10.3390/antiox12040795_

Round 1

Reviewer 1 Report

The paper “Preeclampsia affects lipid metabolism and HDL function in 2 mothers and their offspring” by Stadler, et al. describes that hyperlipidemia is involved in the pathogenesis of preeclampsia and also related to the occurrence of future lifestyle-related diseases. Normal pregnant women are known to have hyperlipidemia, which is further exacerbated by preeclampsia; however, fat is originally important for maternal energy and is physiologically increased. Furthermore, we do not know whether the hyperlipidemia is important for the fetus when the fetus is unable to metabolize lipid. This manuscript contains very useful data. However, this data is simplistic and does not allow the evaluation of the effects of lipid metabolism on cellular functions and organs failure. Additional data is needed to allow evaluation of effects. The interpretation of the data in this manuscript is also often overstated and needs to be rewritten. All concerns listed below, and the discussion should be rewritten to be able to interpret data accurately for improving the overall quality.

The results describe the differences in fat distribution, and although the differences between PE and normal pregnancy were observed, the significance of the differences varied by parameters, and there was little difference between the early-onset and late-onset types, and no significant difference in lipids between the early-onset and late-onset types was observed. If there is no difference between these two types with widely different disease types, it is difficult to say that these results have a significant impact on the pathogenesis of PE.

The functional or metabolic implications of the data have not been evaluated, and there is no information to speculate on the impact of lipid metabolism abnormalities on the pathogenesis of PE. Furthermore, we cannot understand whether these changes are related to future lifestyle-related diseases.

The pathophysiology of hyperlipidemia in pregnancy is very different from that of atherosclerosis in old persons, which is formed by long-term effects, because even normal pregnant women are hyperlipidemic to begin with. Therefore, even though there is a significant difference in the results, we cannot understand whether this degree of hyperlipidemia really affect the pathogenesis of PE. Moreover, if there is no difference in small LDL, the effect of hyperlipidemia is questionable.

There are too many differences between maternal blood and cord blood to understand the relevance. Since the fetus is originally incapable of lipid metabolism, it is not clear whether these changes are truly of fetal origin or maternal origin and furthermore, whether the change is meaningful or not.

Conclusion.

"Early-onset PE lead to a shift in HDL subclass distribution from large to smaller particles, associated with an increase in plasma anti oxidative capacity." is a compensatory response? If so, what is the essence of this response?

"our results suggest that early-onset as well as late-onset PE affect maternal and neonatal lipid metabolism, which may contribute to disease manifestation and to the increased cardiovascular risk later in life.” Isn't this an overstatement?

Author Response

Reviewer 1:

The paper “Preeclampsia affects lipid metabolism and HDL function in 2 mothers and their offspring” by Stadler, et al. describes that hyperlipidemia is involved in the pathogenesis of preeclampsia and also related to the occurrence of future lifestyle-related diseases. Normal pregnant women are known to have hyperlipidemia, which is further exacerbated by preeclampsia; however, fat is originally important for maternal energy and is physiologically increased. Furthermore, we do not know whether the hyperlipidemia is important for the fetus when the fetus is unable to metabolize lipid. This manuscript contains very useful data. However, this data is simplistic and does not allow the evaluation of the effects of lipid metabolism on cellular functions and organs failure. Additional data is needed to allow evaluation of effects. The interpretation of the data in this manuscript is also often overstated and needs to be rewritten. All concerns listed below, and the discussion should be rewritten to be able to interpret data accurately for improving the overall quality.

 Answer: We thank the reviewer for reviewing our manuscript.

Reviewer 1:The results describe the differences in fat distribution, and although the differences between PE and normal pregnancy were observed, the significance of the differences varied by parameters, and there was little difference between the early-onset and late-onset types, and no significant difference in lipids between the early-onset and late-onset types was observed. If there is no difference between these two types with widely different disease types, it is difficult to say that these results have a significant impact on the pathogenesis of PE.

Answer: We would like to apologize if we have not described the objectives of our study in sufficient detail, which may have led to misunderstandings. We would like to kindly point out that it was not a focus of our study to assess differences in fat distribution. We would like to emphasize that our study is an exploratory study that, due to the limited design, is only correlative in nature and does not allow for causal conclusions. Therefore, it was not the aim of our study and we do not in any way describe a causal relationship between changes in lipid metabolism and the occurrence of preeclampsia or an increased risk of cardiovascular disease later in life. We have highlighted this in the Limitation Section and included the following sentence (line 472) “Moreover, our study is exploratory and due to the design limited to be correlative in nature, not permitting causal inference”.

The aim of our exploratory research was to gain a deeper understanding of PE associated alterations on maternal and fetal lipid metabolism, with a particular focus on high-density lipoproteins (HDL). This is of particular importance, given that recent studies have shown that certain subclasses of lipoproteins, especially HDL, are involved in a number of important physiological functions, many of which are important for a healthy pregnancy. HDL is now valued for its role in regulating lipid metabolism, haemostasis, immune response, inflammation and vitamin transport [doi:10.1194/jlr.R035725]. HDL particles exhibit cardio-protective properties and show anti- anti- oxidative effects, including regulation of endothelial functions by promoting nitric oxide production and maintaining endothelial integrity. We believe assessing HDL functional properties and lipid metabolism in mothers affected by PE and their offspring is an important research question. Based on our findings in this study, we hypothesize that the multiplicity of changes in lipid metabolism and HDL particle functionality may be related to disease manifestation and increased cardiovascular risk later in life. However, it certainly requires much larger studies designed to draw firm conclusions.

Reviewer 1: The functional or metabolic implications of the data have not been evaluated, and there is no information to speculate on the impact of lipid metabolism abnormalities on the pathogenesis of PE. Furthermore, we cannot understand whether these changes are related to future lifestyle-related diseases.

The pathophysiology of hyperlipidemia in pregnancy is very different from that of atherosclerosis in old persons, which is formed by long-term effects, because even normal pregnant women are hyperlipidemic to begin with. Therefore, even though there is a significant difference in the results, we cannot understand whether this degree of hyperlipidemia really affect the pathogenesis of PE. Moreover, if there is no difference in small LDL, the effect of hyperlipidemia is questionable.

Answer: We would like to emphasize again, that our study is an explorative study, limited to be correlative in nature, not permitting causal inference. Therefore, we agree with the reviewer that it is not feasible to state whether the found changes in lipid metabolism are directly related to future cardiovascular events. However, we would like to kindly point out that strong evidence is accumulating that hypertriglyceridaemia is associated with endothelial dysfunction. We have discussed that in our manuscript (line 379 – 385). Experimental studies (PMID: 19111829, PMID: 23868936) as well as clinical studies (PMID: 27065244) observed that high plasma triglycerides correlate with endothelial dysfunction, which is associated with cardiovascular risk factors and an independent predictor of cardiovascular events (PMID: 15668353, PMID: 26975705). Therefore, at least in our opinion, it can be assumed that increased plasma triglycerides together with increased oxidative stress in PE lead to an increased burden on the vascular system.

Reviewer 1: There are too many differences between maternal blood and cord blood to understand the relevance. Since the fetus is originally incapable of lipid metabolism, it is not clear whether these changes are truly of fetal origin or maternal origin and furthermore, whether the change is meaningful or not.

Answer: We agree that there are multiple interesting differences between maternal and cord blood, as lipoprotein metabolism in the mother and fetus is not directly linked. We would like to emphasize that the majority of previous studies on lipid metabolism in PE have focused only on affected mothers. One great strength of our study is that we also collected corresponding umbilical cord blood to determine alterations in lipoprotein metabolism and function in the offspring as well. Therefore, we believe that our novel results in cord blood are very important to deepen our understanding that PE affects not only the mothers but also the offspring.

However, as already mentioned, our study is an exploratory study that, due to the limited design, is only correlative in nature and does not allow for causal conclusions. Therefore, we cannot answer whether these changes are actually of fetal or maternal origin and whether they are meaningful beyond that. We agree with the reviewer that further studies designed to answer these research questions are needed.

Reviewer 1: Conclusion.

"Early-onset PE lead to a shift in HDL subclass distribution from large to smaller particles, associated with an increase in plasma anti oxidative capacity." is a compensatory response? If so, what is the essence of this response?

Answer: We have discussed that in our manuscript; maybe the reviewer has overlooked that section (Line 408-420). We write: “This higher anti-oxidative capacity related to hypertension during pregnancy might indicate a compensatory mechanism in response to increased oxidative stress in the circulation. Interestingly, we observed that plasma anti-oxidative capacity correlated with the PE-associated shift from large HDL to small HDL subclasses in early-onset PE. This might be explained, at least in part, by the fact that small, dense HDL are known to exhibit potent antioxidant activity, which may arise from synergy in inactivation of oxidized lipids by enzymatic and nonenzymatic mechanisms [74]. Moreover, we found a significant correlation between uric acid [54] and anti-oxidative capacity. Plasma uric acid is commonly elevated in subjects with impaired kidney function [54] and is a powerful antioxidant and scavenger of singlet oxygen and radicals [55,75]. Uric acid and other hydrophilic antioxidants could explain the relationship between plasma antioxidant capacity and markers of renal dysfunction [76]. However further studies are warranted to underline our, so far speculative, hypothesis”. We hope that our discussion is detailed enough for the reviewer.

Reviewer 1:"our results suggest that early-onset as well as late-onset PE affect maternal and neonatal lipid metabolism, which may contribute to disease manifestation and to the increased cardiovascular risk later in life.” Isn't this an overstatement?

Answer: Following the reviewer's comment, we have tried to formulate our conclusion more cautiously. We write in the revised abstract and conclusion (line 487) “our results suggest that early-onset as well as late-onset PE affect maternal and neonatal lipid metabolism, potentially contributing to disease manifestation and increased cardiovascular risk later in life”. We have added a further sentence to the conclusion (line 489): “However, larger studies are needed to confirm our results and to investigate whether these changes persist after birth of the child”.

Reviewer 2 Report

the manuscript is interesting, well illustrated and generally well written. 

In this study, authors investigated the effects of PE on maternal and neonatal lipid metabolism and parameters of HDL composition and function.

Introduction: a short introduction on preeclamptic placenta alterations deserve to be added. In particular, it deserves to be pointed out that preeclamptic placentas are characterized by the trophoblast immaturity (PMID: 32529396) and vascular dysfunction (PMID: 34831277). This leads to oxidative stress and inflammation that can contribute to, at least some, of the alteration found by the authors (e.g. Sflt-1 and PlGF levels).

2. Material & Methods: Material & Methods must be full described and product codes must be provided

Figures: the number of samples analysed must be added under each figure

Author Response

Reviewer 2:

the manuscript is interesting, well illustrated and generally well written. 

In this study, authors investigated the effects of PE on maternal and neonatal lipid metabolism and parameters of HDL composition and function.

Answer: We are pleased that our manuscript was favorably received by the reviewer and thank the reviewer for the positive comments. 

Reviewer 2: Introduction: a short introduction on preeclamptic placenta alterations deserve to be added. In particular, it deserves to be pointed out that preeclamptic placentas are characterized by the trophoblast immaturity (PMID: 32529396) and vascular dysfunction (PMID: 34831277). This leads to oxidative stress and inflammation that can contribute to, at least some, of the alteration found by the authors (e.g. Sflt-1 and PlGF levels).

Answer: According to the reviewer’s suggestion, we have added a short introduction on alterations in the preeclamptic placenta (Line 50-57).

Reviewer 2: 2. Material & Methods: Material & Methods must be full described and product codes must be provided

Answer: According to the reviewer’s suggestion, we have added more details in the description of Methods and added the catalog numbers of the kits we have used for our measurements.  

Reviewer 2: Figures: the number of samples analysed must be added under each figure

Answer: We thank the reviewer for this comment. We have added the number of analysed samples in each figure legend.

Round 2

Reviewer 1 Report

The quality of papers submitted for consideration includes enough reader's interest and scientific quality. The given paper satisfies requirements for publication of this journal.